# Effect of Heat Treatment and Light Exposure on the Antioxidant Activity of Flavonoids

**Irina Ioannou [1,2,\*], Leila Chekir [3] and Mohamed Ghoul [2]**

[1]   URD Agro-Biotechnologies Industrielles, AgroParisTech, CEBB, 51110 Pomacle, France
[2]   Laboratory of Reactions and Process Engineering (LRGP), Lorraine University, 54505 Vandoeuvre, France; mohamed.ghoul@univ-lorraine.fr
[3]   Laboratory of Cellular and Molecular Biology, Faculty of Dental Medicine, Monastir 5000, Tunisia; leila.chekir@laposte.net
\*   Correspondence: irina.ioannou@agroparistech.fr

**Abstract:** The application of food processes can lead to a modification of both the structure and the activities of flavonoids. In this article, the effect of heat treatment and exposure to light on the antioxidant activity of 6 model flavonoid solutions (rutin, naringin, eriodictyol, mesquitol, luteolin, and luteolin 7-*O*-glucoside) was studied. The evolution of the antioxidant activity measured after heat treatment of 130 °C at 2 h and an exposure to visible light for 2 weeks is measured by the ABTS (2,2′-Azino-bis(3-ethylbenzothiazoline-6-sulfonic acid) diammonium salt) method and represented by a new parameter called ΔTEAC. The model solution of Mesquitol showed the highest increase in ΔTEAC after a heat treatment, a value of 200 mM was obtained. The increase in ΔTEAC is always greater with thermal treatment than with light exposure. Thus, temperature and light lead to different degradation pathways of the flavonoid. In vivo measurements were carried out with solutions of naringin, erodictyol, and luteolin 7-*O*-glucoside. Heated solutions of flavonoids do not exhibit toxicity on cells. The specific activities of superoxide dismutase and glutathione peroxide have been determined and have shown an increased impact on the potential anti-cancer of these solutions by enhancing their cellular antioxidant activity, as well as modulation of the oxidative stress.

**Keywords:** TEAC; flavonoids; heat process; light exposure; degradation products; bioactivities

---

## 1. Introduction

For many years, secondary metabolites from plants such as flavonoids have been acclaimed as natural antioxidant molecules. Indeed, these molecules have many biological activities, giving them a beneficial effect on health. Many studies have shown the link between the consumption of flavonoid-rich products and the prevention of diseases such as cardiovascular and neuro-degenerative diseases [1]. These effects are mainly due to the antioxidant properties that depend on the presence and the localization of functional groups on the flavonoids skeleton [2]. The activity of antioxidants is linked to their capacity to scavenge radical species. The configuration and the number of phenoxy groups are an important parameter to improve antioxidant activity. A catechol structure (phenols in position 3′ and 4′) and an enol group in position 3 (Figure 1) lead to a better free radicals scavenging. Moreover, the presence of an enone moiety contributes to increase further antioxidant properties.

However, the consumption of flavonoid-rich products is mainly done through the consumption of processed food products that are issued from the transformation of raw materials by formulation processes and have been mixed with other ingredients such as tomato sauce or vegetable soup. Flavonoids are sensitive to their environmental conditions (temperature, light, oxygen, and pH) [3]. Thus, the application of formulation processes can lead to a change of the flavonoid structure and,

therefore, on their antioxidant activities. The study of the evolution of the flavonoid antioxidant activity during a change in temperature or light conditions seems relevant to determine the impact of the formulation processes on the flavonoid bioactivities. The evolution of the antioxidant activity of phenolic compounds, after a heat treatment or a storage under light conditions, has been little studied. The major part of the studies deals with phenolic compounds contained in food matrices, thus the antioxidant activity monitored is that of the food raw materials (Tables 1 and 2). Studying the antioxidant activity of phenolic compounds in their food matrices is not objective. Indeed, potential interactions with other molecules might modify the antioxidant activity of the food raw material. Moreover, in these studies, different process conditions (e.g., way to apply the heat treatment, temperature, time) were performed, thus leading to results impossible to compare.

**Figure 1.** The main chemical features responsible for antioxidant activity.

**Table 1.** Review on the studies dealing with the evolution of antioxidant activity (AA) of phenolic compounds (PC) during a heat treatment (HT).

| Food/Model Solution of Flavonoid | Process Parameters | Antioxidant Activity | References |
|---|---|---|---|
| Red grape, pomace peel | Drying (140 °C; 3 h) | Decrease of the AA (50%) for a decrease of PC (32.6%) | [4] |
| Onion bulbs | Boiling (60 min) | Decrease of the AA | [5] |
| Citrus peel | HT (100 °C; 60 min) | 13% increase for AA for an increase of the PC of 43% | [6] |
| Rutin/Luteolin 7-*O*-glucoside model solutions | HT (100 °C, 360 min) | Loss of 15% of the AA | [7] |
| Quercetin model solution | HT (100 °C) | AA constant | [8] |
| Shiitake mushroom | HT (121 °C, 30 min) | Increase of the AA by 2.0-fold | [9] |
| Sweet potato | Steaming 40 min | Increase of the AA | [10] |
| Oak corn | HT (200 °C; 10 min) | Increase of the AA | [11] |
| Red and green peppers | Frying (180 °C; 5 min) Microwave (500 W; 5 min) | AA constant for a PC content unchanged | [12] |
| Garlic/Onion | HT (100 °C; 10 min) | Decrease of the AA | [13] |
| Strawberry juice | HT (90 °C; 30/60 s) | AA constant | [14] |
| Celery | Immersion treatment (55 °C, 60 s) Storage 24 days at 0 °C | AA constant Decrease until 6 h then an increase until 15 h | [15] |
| Yellow and Black soybeans | HT (100 °C, 60 min) | Decrease of the AA | [16] |
| Bilberry | Drying (80 °C; 30 min/125 °C; 10 min) | Increase of the AA | [17] |
| Beans | Cooking (100/121 °C) | Increase of the AA and the PC | [18] |
| Buckwheat seeds and groats | Roasting (160 °C; 30 min) | Decrease of the AA | [19] |
| Flour of tartary buckwheat | Roasting (120 °C; 40 min) Microwave (700 W; 10 min) | Decrease of the AA | [20] |
| Small black soybean | HT (250 °C; 30 min) | Increase of the AA | [21] |
| Little millet | HT (165 °C; 75 s) | Increase of the AA | [22] |
| Blackberry juice | HT (90 °C; 5 h) | Decrease of the AA | [23] |
| Green leaves of leeks | Steaming (30 min) | Increase of the AA 20% for a decrease of PC 20% | [24] |
| Kumquat | 130 °C, 30 min | Increase of the AA | [25] |
| Sorgo | Baking (121 °C; 25 min) | Increase of the AA 16% for a decrease of PC 10% | [26] |
| Cardon | Frying in sunflower oil (115 °C; 10 min) | Increase of the AA 20% for an increase of PC 20% | [27] |

Herein, we propose to establish the effect of a thermal treatment and an exposure to light on the evolution of the antioxidant activity and the biological activities of six flavonoid model solutions.

First of all, a new parameter is proposed to follow the evolution of the antioxidant activity. In a second part, the evolution of the antioxidant activity of flavonoid model solutions (naringin (NG), eriodictyol (Erio), rutin, mesquitol, luteolin, and luteolin 7-*O*-glucoside (L7G)) during the application of an isothermal treatment and an exposure to light are presented and compared. Different chemical structures were studied in order to establish structure-activity relationships (SARs). In a third part, thermal processing, such as the one undergone during transformation steps of raw material to finished products, was studied on some flavonoids to evaluate its impact on their potential anti-cancer capacity by enhancing their cellular antioxidant activity, as well as modulation of the oxidative stress. The specific activities of superoxide dismutase and glutathione peroxide, as well as malondialdehyde contents, were determined.

**Table 2.** Review on the studies dealing with the evolution of antioxidant activity during a storage under light conditions.

| Food/Model Solution of Flavonoid | Process Parameters | Antioxidant Activity | References |
|---|---|---|---|
| Pomegranate | 5 days, 5 °C,UV-C | Constant | [28] |
| Blueberry | 4 days, 24 °C, light | Increase | [29] |
| Mulberry | 10 h, fluorescent light | Increase | [30] |
| Mushroom | 15 days, 4 °C, pulsed light | Decrease (28 J/cm$^2$) and constant (4.8 J/cm$^2$) | [31] |
| Blueberry | 28 days, 0 °C, UV light 6 kJ/m$^2$ | Increase | [32] |
| Paprika | 7 days, 8 °C, UV-C 15 kJ | Constant | [33] |
| Tomato | 21 days, 20 °C, UV-C | Increase | [34] |
| Apple, tomato, bell pepper | 7 days, 7 °C, LED (590 nm) | Increase | [35] |
| *Piper betle* | 6 months, 5/25 °C, light | Decrease | [36] |
| Soybean sprouts | LED and FIR light | Increase | [37] |

## 2. Materials and Methods

### 2.1. Chemicals

Naringin (purity > 95%), rutin (purity > 95%), 2'-azinobis (3-ethyl-benzothiazoline-6-sulfonic acid), and diammonium salt (ABTS) were purchased from Sigma-Aldrich Chemical (Lyon, France). Eriodictyol, luteolin, luteolin 7-*O*-glucoside (purity > 98%) were purchased from Extrasynthese (Lyon, France). Mesquitol (purity > 90%) was extracted from a Kenya tree Proposis juliflora [38]. Trolox and potassium persulfate were purchased from Fluka (Lyon, France). Methanol and ethanol were from Carlo Erba (Marseille, France) and VWR (Paris, France), respectively. All reagents and solvents were of analytical grade.

### 2.2. Model Solutions of Flavonoids

The flavonoids chosen for this study belong to different classes of flavonoids (flavonols, flavanones, flavanols, and flavones). They differ by the number of phenol groups, their position on the main aromatic rings, the presence of an enone structure and a carbonyl function on the B cycle. Table 3 presents the different chemical features of the flavonoids chosen.

**Table 3.** Structural features of the flavonoids studied.

| Flavonoid | Acronym | Class | Glycosylation | Enone Structure | OH Groups | Phenol Position |
|---|---|---|---|---|---|---|
| Rutin | | Flavonol | Glucose-Rhamnose in position 3 | Yes | 4 | 4'-5'-5-7 |
| Narangin | NG | Flavanone | Glucose-Rhamnose in position 7 | No | 2 | 4'-5 |
| Eriodictyol | Erio | Flavanone | No | No | 4 | 3'-4'-5-7 |
| Mesquitol | | Flavanol | No | No | 5 | 3'-4'-3-7-8 |
| Luteolin | | Flavone | No | Yes | 4 | 3'-4'-5-7 |
| Luteolin 7-*O*-glucoside | L7G | Flavone | Glucose in position 7 | Yes | 3 | 3'-4'-5 |

Aqueous solutions of six flavonoids were prepared according to their solubility limit (naringin 0.29 mM, rutin 0.041 mM, mesquitol 0.93 mM, eriodictyol 0.059 mM, luteolin 0.0087 mM,

luteolin 7-*O*-glucoside 0.014 mM). Flavonoids were solubilized in aqueous buffer in the dark at 25 °C until maximum solubilization was achieved. The pH value of the different solutions is equal to 6.6 ± 0.1.

The antioxidant activities in vitro and in vivo were assessed from model solutions of flavonoids treated at a temperature of 130 °C during 2 h [39,40].

## 2.3. Heat Treatment

A heat treatment was applied to model solutions of flavonoids in an isothermal way. The solutions were immersed in an oil bath W8518D (Huber, Bergheim, Germany) at different temperatures for two hours. 10 mL samples of each solution were put into screw-cap pyrex tubes and heated from 30 to 130 °C. Samples were taken every 15 min and cooled in an ice bath to 30 °C [41]. Studying temperatures above 130 °C did not seem relevant as they were not very representative of the temperatures used in food processes.

## 2.4. Light Exposure

10 mL samples of each model solution were put in screw-cap pyrex tubes and exposed to light in a climatic cell Climacell (Thermofisher, Paris, France) to mimic the visible light. Oxygen concentration of the solutions was 0.19 mM and climatic cell was set at 25 °C. Model solutions were exposed to 0% light (darkness) and 100% light. An intensity of 100% is equivalent to an exposure at 16.5 klux. Each day, over a 15-day period, one sample was collected [42].

## 2.5. Measurement of Flavonoid Content and Antioxidant Activity

### 2.5.1. HPLC/DAD Analysis

Flavonoid contents were analyzed with an Elite Lachrom HPLC system (VWR HITACHI, Paris, France) which consisted of a quaternary pump (L-2130), an autosampler (L-2200), and a diode-array detector (L-2455). The analyses were carried put on a C18 column (150 × 4.6 mm) with 2.6 μm particle size (Grace). The different compounds were separated using a water (A) + methanol (B) elution system. Samples were eluted at a flow rate of 1 mL/min by a gradient of: 95% water (A) at time 0 up to 100% (B) at 10 min and finally 95% (A) at 20 min before the end of the program. The total duration is 30 min. The injection volume was of 50 μL and the column temperature was set at 50 °C. The detection by DAD was performed simultaneously at 200, 280, 254, and 350 nm. Results are expressed as a degree of flavonoid degraded expressed in percentage as described by Equation (1).

$$\text{Degree of flavonoid degraded } (\%) = 100 \times (C_t / C_0) \qquad (1)$$

With $C_t$, the flavonoid content at t min of the process and $C_0$, the initial flavonoid content.

### 2.5.2. ABTS$^{+\bullet}$ Method

Antioxidant activity was determined by the ABTS assay according to the method from Re et al. (1999) with modification for use in microplates [43]. A stable standard solution of ABTS$^{\bullet+}$ was prepared by reacting 7 mM/L of an aqueous solution of ABTS with 2.45 mM/L of potassium persulfate. After the addition of 220 μL of the standard solution ABTS$^{\bullet+}$ to 80 μL of flavonoid model solution, the absorbance at 734 nm is determined at 30 °C, exactly 1 min after the initial stirring ($A_0$) and at 15 min ($A_t$) using a spectrofluorimeter equipped with a 96-well polystyrene plate (SAFAS flx Xenius, Monaco). The percentage of inhibition (PI) was calculated between $A_0$ and $A_t$.

The slope of the linear regression PI = f (concentration of the extract μM) is divided by the slope of the linear regression PI = f (concentration of Trolox μM). The value found indicates the amount of trolox needed to obtain inhibitory power equivalent to one mole of the sample analyzed. Results were expressed as Trolox Equivalent Antioxidant Capacity (TEAC) in mM Trolox equivalent.

### 2.6. Assessment of the In Vivo Bioactivities

#### 2.6.1. Animals

Male Balb/c mice (20–22 g) were purchased from Pasteur Institute (Tunis, Tunisia). Animals were housed under standard conditions of temperature, humidity, and light (24 h cycle of 12 h light/12 h dark). They were fed with a commercial pellet diet and water ad libitum. All experiments were performed in accordance with the guidelines for the care and use of laboratory animals as published by the National Institute of Health. All experiments obtained the explicit approval of the Tunisian Ethics Animal Committee.

#### 2.6.2. Preparation of Primary Splenocytes and Macrophages

Spleen mice lymphocytes were obtained as previously reported [44,45]. After washing with phosphate buffered saline (PBS, pH 7.4), cells were resuspended in complete RPMI medium (GIBCO, BRL) containing 10% fetal bovine serum (FBS; GIBCO) and 100 mg/mL gentamycin (Gibco-BRL, Paisley, UK). Other mice were used to provide peritoneal macrophages as previously described [46]. Cell viability was assessed using the trypan blue exclusion technique.

#### 2.6.3. Preparation of the Liver, Brain, Spleen, and Kidney Extracts

The liver, kidney, spleen, and brain, of mice treated with luteolin 7-*O*-glucoside (L7G) and heated luteolin 7-*O*-glucoside (hL7G) or control (10 mg/kg), were homogenized in the presence of PBS and centrifuged at 12,000 rpm for 30 min at 4 °C. The supernatant was collected, aliquoted, and stored at −80 °C until use. The total protein concentration was determined following Bio-Rad protein assay with bovine serum albumin used as standard [47].

#### 2.6.4. Cell Treatment

Splenocytes treated with various concentrations of molecules and optimal concentrations of lectin (5 μg/mL) and lipopolysaccharide LPS (5 μg/mL) were added to each well separately for priming T and B cells, respectively. Macrophages ($3 \times 10^5$ cells per well) were incubated with various concentrations of flavanones and with or without the addition of LPS (5 μg/mL) solubilized in complete RPMI-1640 medium [48]. The cells were maintained at 37 °C in a 5% $CO_2$ humidified atmosphere.

#### 2.6.5. Analytical Measurements

1.   T and B cell proliferation assay

Lymphocyte proliferation was assessed by the mitochondrial-dependent reduction of 3-(4, 5-dimethylthiazol-2-yl)-2, 5-diphenyl tetrazolium bromide (MTT) to purple formazan [49]. Splenocyte suspension, in RPMI-1640 medium ($2.5 \times 10^6$ cells/mL; 100 μL aliquot/well), was pre-incubated in 96-well plate for 24 h, before the addition of both mitogen (LPS or Lectin, each at 5 μg/mL) and the tested compound solubilized in RPMI. Cells were then incubated at 37 °C in humidified 5% $CO_2$ atmosphere for an additional 48 h. Thereafter, 40 μL of MTT (5 mg/mL) were added to RPMI solution and incubated for 2 h at 37 °C. Plates were then centrifuged again, the MTT removed from each well and the formazan was dissolved in 100 μL of dimethylsulfoxide (98% DMSO). After incubation at 37 °C for 15 min, absorbance of formazan, formed in each well, was measured at 570 nm in a microplate reader (Thermo Scientific, Vantaa, Finland). The percentage of proliferation was ultimately calculated using Equation (2) [48].

$$\text{Proliferation } (\%) \ = \ 100 \times (A_s - A_C)/A_C \qquad (2)$$

2.   Evaluation of lipid peroxidation status

Lipid peroxidation was evaluated indirectly by monitoring the production of malondialdehyde (MDA) in the brain, spleen, renal, and liver extracts, employing the thiobarbituric acid reactive

substances assay [50]. Briefly, extracts of different organs (50 µL) were mixed with 1 mL of trichloroacetic acid (20%), and 1 mL of thiobarbituric acid (TBA) (0.67%) was freshly prepared. The samples were subsequently incubated at 95 °C for 30 min and then at 25 °C for 60 min. After sample centrifugation, TBA reacts with the oxidative degradation products, generating red complexes absorbed at 530 nm as measured using in a microplate reader (Thermo Scientific, Vantaa, Finland). The concentration of MDA (nanomoles per milligram proteins) was obtained by extrapolating absorbance values to concentrations of standard curve of MDA.

3. Determination of reduced glutathione

The concentrations of GSH were determined based on GSH oxidation with DTNB (5, 5′-dithiobis-2-nitrobenzoic acid). GSH levels were determined as described by Arjumand and Sultana (2011) [51]. GSH content was expressed as nM/mg of protein. Analysis of glutathione peroxidase activity. The activity of glutathione peroxidase (GPx) was performed from the extracts of different organs following Flohe and Gunzler (1984) [52]. GPx activity was expressed as units/mg of protein.

4. Measurement of catalase activity

Catalase (CAT) activity was evaluated by measuring the proportion of decrease in 10 mmol/L hydrogen peroxide ($H_2O_2$) absorbance at 240 nm. The activity was estimated in terms of µmol $H_2O_2$ consumed/min/mg of protein [53].

5. Analysis of superoxide dismutase activity

The superoxide dismutase (SOD) activity was determined in the extracts of different organs by the inhibition of Nitro Blue Tetrazolium (NBT) reduction assay [54,55]. The reaction mixture consisted of 50 µL of the liver, brain or kidney extracts, 2 mM NBT, 10 mM methionine, 2.4 mM riboflavin, and 0.1 mM EDTA in a final volume of 1.7 mL. The reaction was carried out for 15 min illuminated with a UV fluorescent lamp. The absorbance was then measured at 560 nm. An enzymatic unit was defined as the amount of cytosol required to inhibit 50% of the reaction without enzyme.

6. Cellular anti-oxidant activity (CAA) assay

A cellular anti-oxidant activity (CAA) assay, developed by Wolfe and Liu [56], was employed to measure the anti-oxidant potential of the tested samples. Briefly, splenocytes and macrophages were seeded at a density of $5 \times 10^5$ and $6 \times 10^4$, respectively (in 100 µL PBS). Triplicate wells were then treated with 10 µL of each sample (concentrations ranging from 21.6 µg/mL to 2.7 µg/mL) along with 5 µL of a 25 µM solution of 2′, 7′-dichlorofluorescin diacetate (DCFH-DA; Fluka, Steinheim, Germany). After 1 h of incubation, 100 µL aliquot of 600 µM 1.2 mM 2, 2′-azobis (2-amidinopropane) dihydrochloride (ABAP) (Sigma Aldrich, Steinheim, Germany) in PBS, was applied to the cells. In this method, DCFH-DA is taken up by cells and deacetylated to DCFH (2′, 7′-dichlorofluorescin). Peroxyl radicals generated from ABAP lead to the oxidation of DCFH to fluorescent DCF. Accordingly, cells treated with natural compounds that have any anti-oxidant activity should have lower fluorescence compared to untreated cells. Fluorescence of each well, was followed every 5 min during 1 h using a fluorescence microplate reader (Biotek, Winooski, USA) with 538 nm emission and 485 nm excitation filters. Each plate included triplicate control wells containing cells treated with DCFH-DA and the oxidant ABAP and blank wells containing cells with PBS but without oxidant ABAP. Fluorescence values for the blank controls and initial fluorescence values were subtracted from the sample fluorescence values. The area under the fluorescence vs. time curve was integrated at each time point to calculate the CAA units using Equation (3) with $\int SA$ is the integrated area under the sample fluorescence versus time curve, and $\int CA$ is the integrated area under the control fluorescence versus time curve.

$$CAA \text{ unit } = 100 - \left( \int SA / \int CA \right) \times 100 \tag{3}$$

### 2.7. Statistical Analysis

Experiments of heat treatment and light exposure were performed using three replicates. Statistical analysis was performed by using the freeware R (2.11.1). In all the tests, criteria for statistical significance was $p < 0.05$.

For the assessment of in vivo bioactivities, results are mean ± standard deviation on five replicates. The statistical significance of results was evaluated by a two-way ANOVA followed by the Tukey-multiple comparison test. For Figures 6 and 7 and Table 4, following notations were used: * $p < 0.05$, means significant difference between treated animals and control group (TN). ** $p < 0.01$, means significant difference between treated animals and control group (TN). *** $p < 0.001$, means significant difference between treated animals and control group (TN). ♦ $p < 0.05$ means significant difference between animals treated with NG and hNG.

## 3. Results

### 3.1. ΔTEAC a New Parameter to Understand the Evolution of the Antioxidant Activity of the Flavonoids

Monitoring the TEAC (Trolox Equivalent Antioxidant Capacity), measured by the ABTS method, according to the treatment time is not sufficient to understand the evolution of the antioxidant activity. Indeed, the antioxidant activity of the solution treated depends on the residual content of the flavonoid studied. Figure 2 presents the percentage of degraded rutin and the evolution of its antioxidant activity according to the heating time.

Figure 2a shows that rutin is sensitive to high temperatures. At a temperature of 70 °C, less than 10% is degraded after two hours of treatment. The degradation is faster with an increase of the temperature. At 130 °C, a total degradation is reached after 30 min. The complete analysis of kinetic data was previously realized [41].

On Figure 2b, the evolution of the TEAC during two hours is given for the different temperature. It shows that the evolution of the TEAC of the solution is different according to the temperature. For temperatures below 100 °C, the TEAC is constant or increases whereas for temperature higher than 100 °C the TEAC decreases until a threshold despite the total degradation of the rutin. Thus, we notice that the TEAC is not correlated to the rutin content. It would indicate that antioxidant activity of the solution treated is not only induced by the rutin content, but also by either the degradation products generated during the treatment or the effect of the synergies between molecules in solution. To analyze the effect of the degradation of rutin on the evolution of its antioxidant activity, we propose to determine only the TEAC induced by the degradation products of rutin.

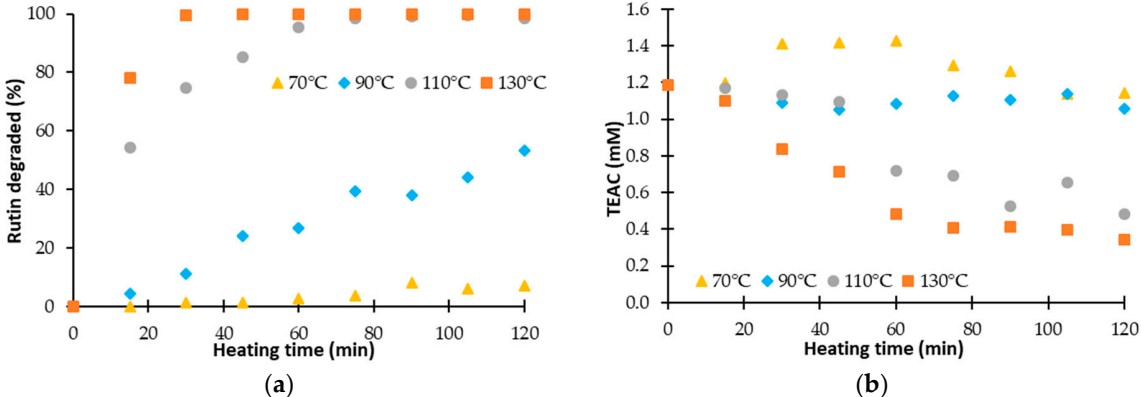

(a)        (b)

**Figure 2.** (**a**) Percentage of degraded rutin during a heat treatment and (**b**) evolution of the Trolox Equivalent Antioxidant Capacity (TEAC).

The value of the TEAC expresses the global antioxidant capacity of the solution including the antioxidant capacity of the flavonoid studied and that of their degradation products. We propose

to calculate ΔTEAC, which corresponds to the TEAC induced only by the degradation products of rutin. For this, the difference is calculated between the global antioxidant capacity and the antioxidant capacity predicted according to the residual content of the flavonoid (Equations (4) and (5)).

$$TEAC' = \frac{TEAC_0}{C_0} \tag{4}$$

With $TEAC_0$ the antioxidant capacity of the flavonoid before the application of the process ($t_0$) and $C_0$ the initial concentration in flavonoid. The TEAC corresponds to a specific activity to a flavonoid given.

The ΔTEAC is calculated according to Equation (5):

$$\Delta TEAC = TEAC_t - TEAC' \times C_t \tag{5}$$

With $TEAC_t$ the antioxidant capacity of the flavonoid solution after t minutes of processing and $C_t$ the residual content in flavonoid at the same time t.

If the ΔTEAC is positive, it means that in solution, there are molecules other than the native flavonoid that have an antioxidant activity or that induce a positive synergistic effect. These molecules are degradation products of the native flavonoid. If the ΔTEAC is negative, it means that all degradation products have an overall pro-oxidative activity or cause negative synergies between the molecules present in solution.

### 3.2. Effect of a Thermal Treatment and an Exposure to Light on the Evolution of ΔTEAC

The evolution of ΔTEAC for a heat treatment of 130 °C during two hours and for an exposure to light during two weeks are presented for the flavanones (naringin and eriodictyol), the flavones (luteolin and luteolin 7-*O*-glucoside) and for rutin and mesquitol, in Figure 3, Figure 4, Figure 5, respectively.

As seen in Figure 3a, the values ΔTEAC for naringin and eriodictyol are positive during a heat treatment at 130 °C, demonstrating that degradation products of these flavanones have an antioxidant activity or induce a positive synergy. The evolution of the ΔTEAC for naringin corresponds to an increase until a threshold around 10 mM. For eriodictyol, an increase is observed at 60 min until 17 mM, then a decrease happens until 15 mM. Thus, the ΔTEAC obtained is higher with eriodictyol than with naringin. However, one cannot determine which solution is the most antioxidant after a heat treatment as the extent of degradation is different between these two molecules (Figure 3c). Eriodictyol is totally degraded after two hours of heat treatment whereas the naringin content is only decreased by 20%. Thus, different proportions of degraded products are present in solution.

In Figure 3b the evolution of ΔTEAC for naringin and eriodictyol is represented during a storage in the dark and during an exposure to light. The ΔTEAC are lower than those during a heat treatment, indicating that degradation products obtained by the effect of the light are different from those obtained by a heat treatment. Standard deviations being equal to 2, no significant evolution can be determined for the model solution of eriodictyol stored in the dark. A positive evolution is noticed for the naringin ΔTEAC with and without light conditions. The percentage of degraded naringin is different in these two conditions (Figure 3d) but both lead to degradation products with an antioxidant activity. A negative evolution is observed for eriodictyol exposed to light. The decomposition of eriodictyol due to a light exposure would lead to pro-oxidant products of degradation [41].

As shown in Figure 4a, a positive ΔTEAC is observed for the two flavones during a thermal treatment. Degradation products of luteolin 7-*O*-glucoside provide a ΔTEAC of 15 whereas for luteolin, values around 4 are obtained. This result could be explained by the degradation pathways of luteolin and luteolin 7-*O*-glucoside. Indeed, after a heat treatment (130 °C), luteolin 7-*O*-glucoside would lead to the formation of luteolin, which possesses an antioxidant activity higher than luteolin 7-*O*-glucoside whereas the degradation products of luteolin would have an antioxidant activity lower than luteolin. Indeed, TEAC of luteolin is equal to 5.56 ± 1.45 whereas that of luteolin 7-*O*-glucoside is equal to 4.21 ± 0.56 [41].

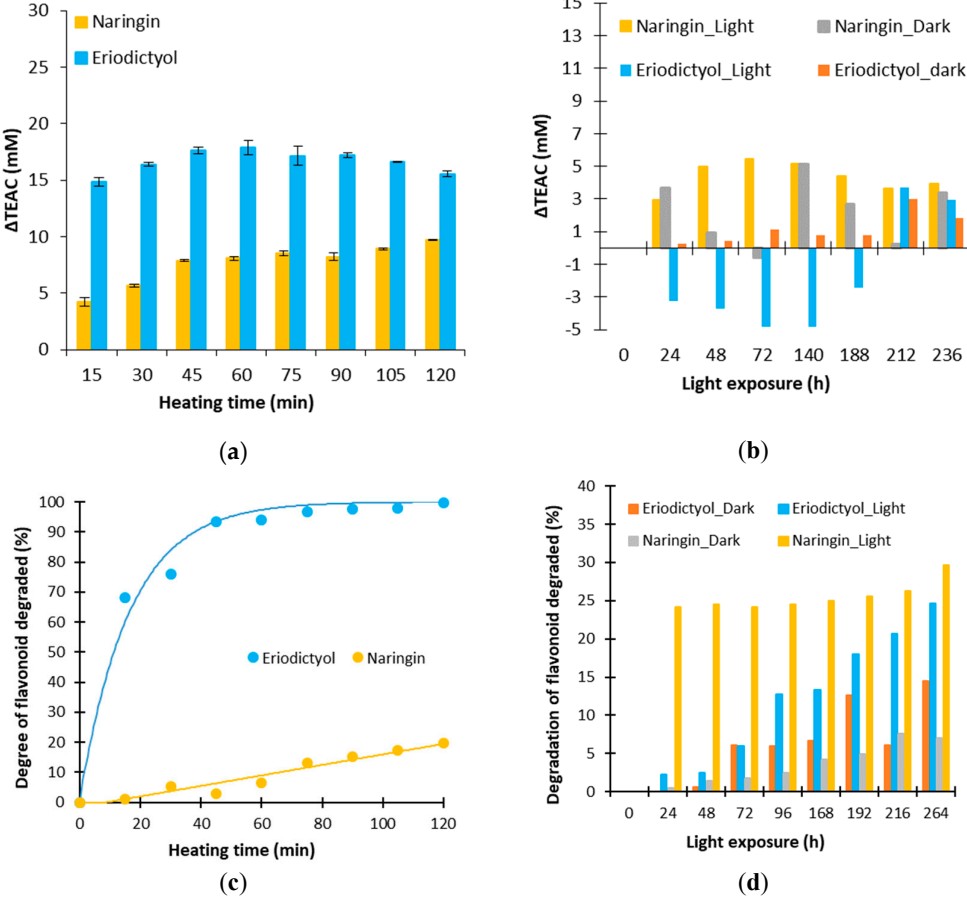

(a)

(b)

(c)

(d)

**Figure 3.** Evolution of ΔTEAC and percentage of flavonoid degraded for naringin and eriodictyol for a thermal treatment of 130 °C (**a**,**c**) and an exposure to light (**b**,**d**).

Figure 4b shows the ΔTEAC of luteolin and luteolin 7-*O*-glucoside stored in the dark and exposed to light. Variations of ΔTEAC are low compared to those obtained after a heat treatment. Standard deviations are between 2 and 3, so a significant decrease was observed only for the luteolin 7-*O*-glucoside exposed to light. Thus, light would induce pro-oxidant products of degradation from a luteolin 7-*O*-glucoside model solution.

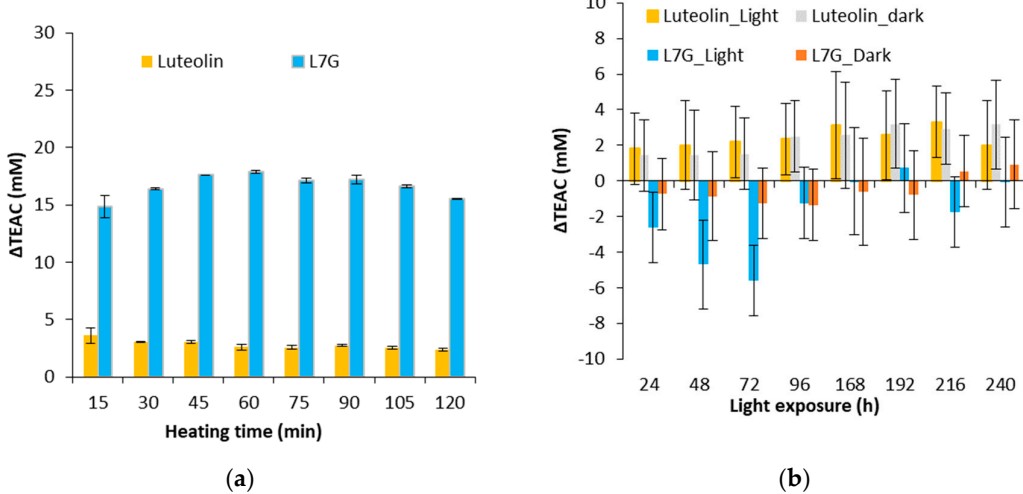

(a)

(b)

**Figure 4.** Evolution of ΔTEAC for flavones (luteolin and luteolin 7-O-glucoside) for a thermal treatment of 130 °C (**a**) and an exposure to light (**b**).

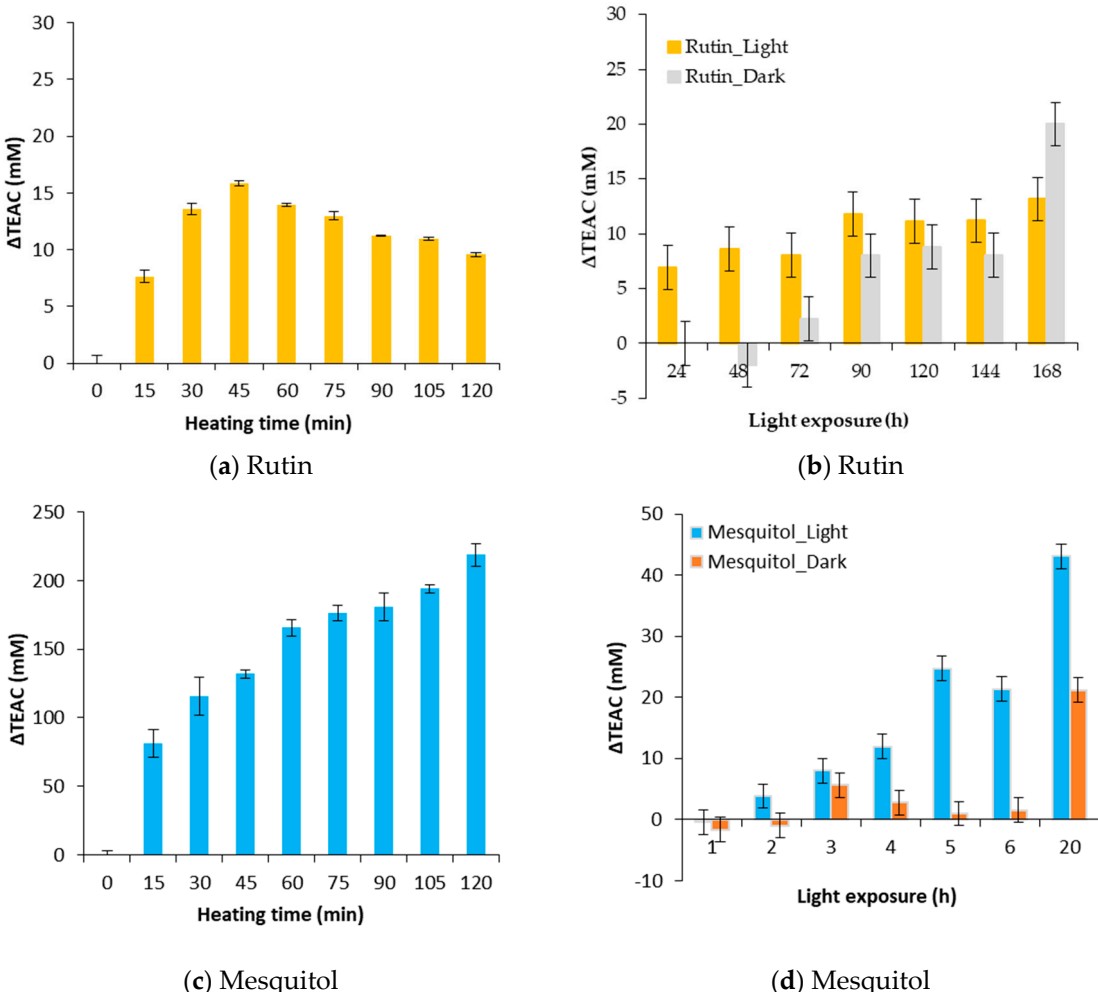

(**a**) Rutin

(**b**) Rutin

(**c**) Mesquitol

(**d**) Mesquitol

**Figure 5.** Evolution of ΔTEAC for a thermal treatment of 130 °C (**a**,**c**) and an exposure to light (**b**,**d**) for rutin (**a**,**b**) and mesquitol (**c**,**d**).

As shown in Figure 5a, positive ΔTEAC is obtained during the heat treatment at 130 °C of rutin and mesquitol model solutions. It should be noted that, for mesquitol, a value of 200 mM is reached after 2 h of treatment, the residual mesquitol being less than 10% in solution. Thus, the degradation products of mesquitol have a very high antioxidant activity. For the rutin, a value of ΔTEAC of 15 mM is obtained at 45 min of heating, this latter decreases down to 10 after two hours of treatment. This result could be explained by the fact that degradation products, appeared within the first 45 min, are themselves degraded by heat to give "second" degradation products.

During a storage in the dark or an exposure to light, ΔTEAC obtained are always lower than those obtained by heat treatment. Values of 13 mM are obtained for the rutin exposed to light and 20 for rutin stored in the dark after 7 days. For mesquitol, value of 40 mM is obtained for an exposition of 20 h in the visible light. Mesquitol is very sensitive to light, after 20 h of exposure, mesquitol is entirely degraded.

### 3.3. Evolution of the Biological Activities of the Flavonoid Model Solutions Thermally Treated

The first set of analysis examines the impact of native molecules and their derivatives issued from thermal treatment on murine splenocyte activation. It appears that splenocyte proliferation increases significantly in the presence of the tested molecules (naringin (NG); luteolin 7-*O*-glucoside (L7G) and eriodictyol (Erio)) (Figure 6). In fact, the degradation products do not exhibit any cytotoxicity compared to native molecules. Heated molecules ensure better lymphocyte proliferation.

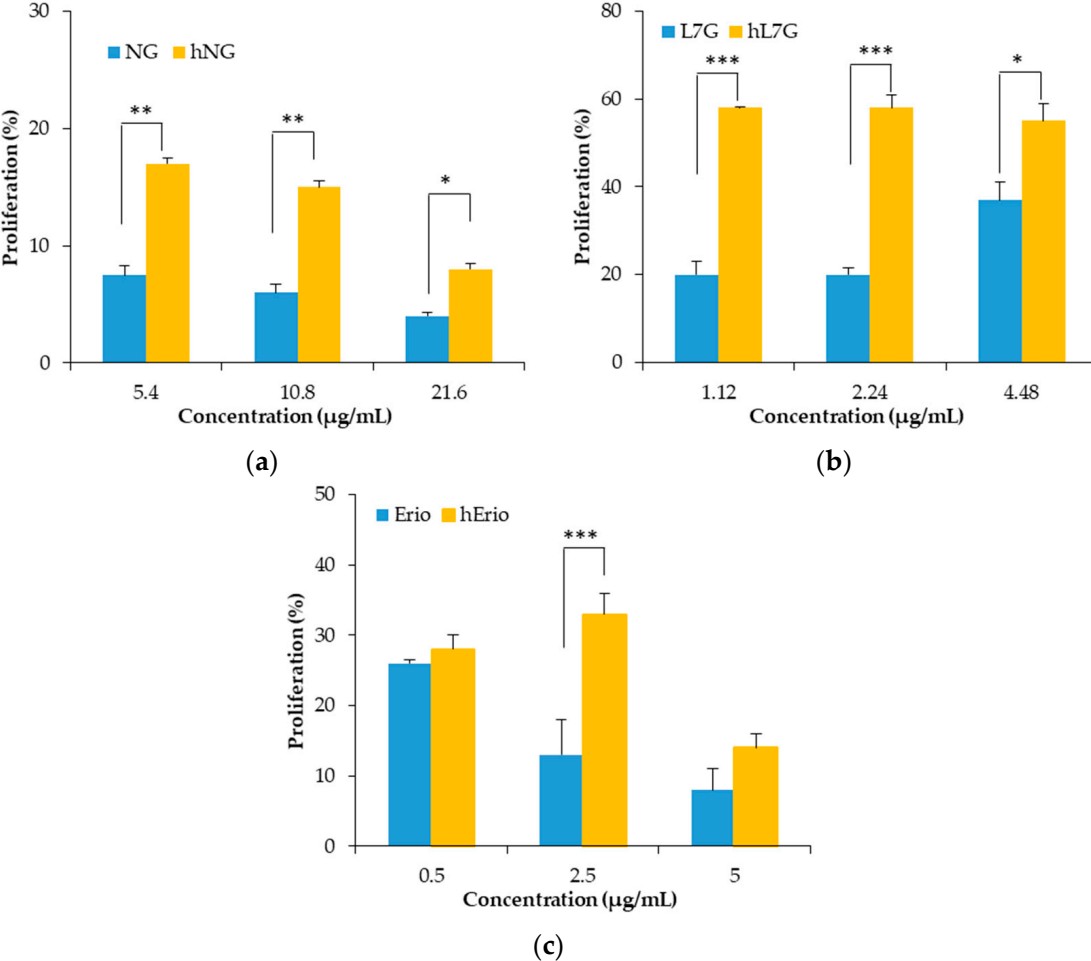

**Figure 6.** Effects of heated flavonoids and native flavonoids on spleen proliferation responses. (**a**) naringin (NG) and heated naringin (hNG), (**b**) luteolin 7-*O*-glucoside (L7G) and heated luteolin 7-*O*-glucoside (hL7G), (**c**) eriodictyol (Erio), and heated eriodictyol (hErio).

On the other hand, heated NG, L7G, and Erio expand the cellular antioxidant activity of splenocytes and macrophages. Native and heated molecules were able to inhibit the oxidation of DCFH.

In macrophages, native L7G and NG showed stronger cellular antioxidant activity (CAA) compared to the heated molecule. At the highest tested concentration, NG inhibits more significantly ABAP-induced DCFH oxidation (39.8%) compared to heated naringin (hNG) (32.17%). Likewise, native and heated L7G inhibits almost equally ABAP-induced DCFH oxidation by 72.81% and 72.32%, respectively. However, heated Erio exhibited stronger antioxidant activity by 78.95% compared to native Erio.

In contrast, in splenocyte, the heated flavonoids prevented ABAP-induced DCFH oxidation in a dose-related fashion. Figure 7 revealed that the oxidation of DCFH-DA by peroxyl radicals, derived from ABAP, was significantly reduced by heated NG, heated L7G and heated Erio, reaching 52.68%, 72.3% and 83.88%, respectively, at the highest tested concentrations. Heated molecules seem to be more efficient in protecting splenocytes against peroxyl radical-induced oxidation compared to native molecules (13.6%, 60%, and 46.5% respectively) [39,40,57]. These findings are confirmed by in vivo assay. It was noted that both native and treated naringin decrease malondialdehyde level in liver, brain, and kidney cells as compared to control. Besides, native and heated naringin treatment significantly enhance the spontaneous superoxide dismutase activity and conserve the GPx activity in mice kidney and brain homogenate (Table 4) [39].

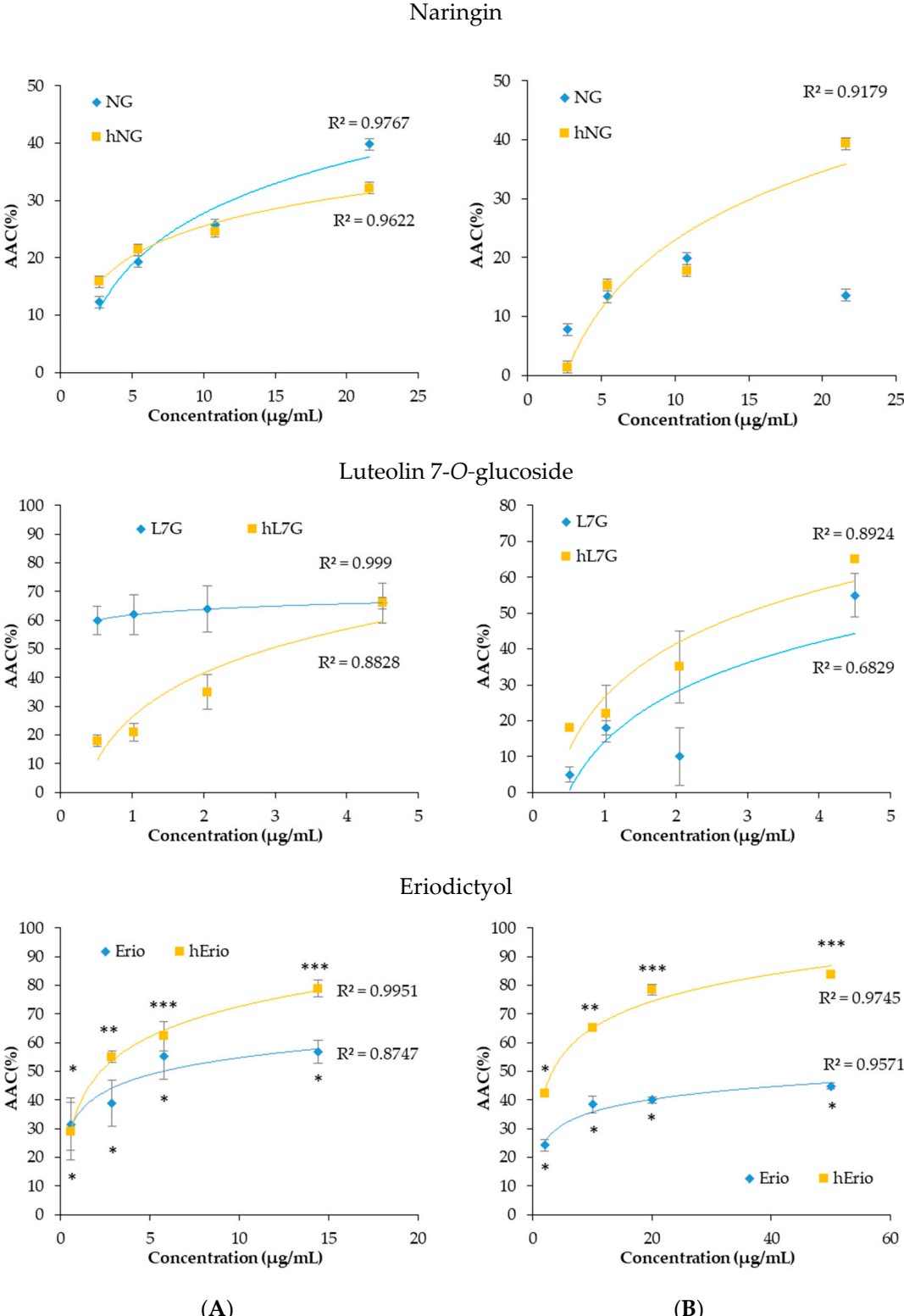

**Figure 7.** Dose-response curves of inhibition of peroxyl radical-induced DCFH oxidation in macrophages ($6 \times 10^4$ cell per well) (**A**) and spleen cells ($5 \times 10^5$ cells per well) (**B**) using a cellular antioxidant activity assay in the presence of heated and native molecules.

**Table 4.** Malondialdehyde (MDA), superoxide dismutase (SOD) and glutathione peroxidase (GPx) levels in liver, kidney, and brain homogenates from mice treated by naringin (NG) or heated naringin (hNG) and control groups (TN).

|  |  | MDA (nmol/mg Protein) | SOD (U/mg Protein) | GPx (U/mg Protein) |
|---|---|---|---|---|
| **Liver** | TN | 2.70 ± 0.29 | 0.730 ± 0.057 | 3.07 ± 0.11 |
|  | NG (40 mg/kg b.w.) | 2.08 ± 0.31 | 0.920 ± 0.081 | 3.24 ± 0.22 |
|  | hNG (40 mg/kg b.w.) | 2.04 ± 0.21 | 0.890 ±.0076 | 3.83 ± 0.16 |
| **Kidney** | TN | 1.93 ± 0.20 | 0.430 ± 0.053 | 4.09 ± 0.19 |
|  | NG (40 mg/kg b.w.) | 1.58 ± 0.19 | 0.650 ± 0.043 * | 3.71 ± 0.21 |
|  | hNG (40 mg/kg b.w.) | 1.97 ± 0.23 | 0.830 ± 0.073 ** | 3.83 ± 0.24 |
| **Brain** | TN | 2.50 ± 0.27 | 0.340 ± 0.012 | 4.01 ± 0.39 |
|  | NG (40 mg/kg b.w.) | 2.05 ± 0.26 | 0.790 ± 0.690 ** | 3.55 ± 0.46 |
|  | hNG (40 mg/kg b.w.) | 1.94 ± 0.19 * | 0.490 ± 0.022 ♦ | 3.40 ± 0.29 |

b.w.: basis weight.

## 4. Discussion

As shown in Tables 3 and 4, the antioxidant activity of food matrices containing phenolic compounds after a heat treatment and an exposure to light can either remain constant, increase, or decrease. The evolution depends on the interactions between molecules and food matrices, and on the operating conditions. Only two studies dealt with the evolution of the antioxidant activity of a model solution of flavonoids. Heating rutin and luteolin 7-*O*-glucoside at 100 °C for 6 h led to a 15% decrease in their antioxidant activity with a decrease in their content of 22% and 16%, respectively [7]. However, the antioxidant activity of a solution of quercetin remain constant after a heat treatment at a temperature of 100 °C for 4 h, although quercetin has completely degraded [8]. Thus, the evolution of the antioxidant activity is not correlated to the content of the flavonoid studied. These results are coherent to our results.

For the same flavonoid, ΔTEAC have a different evolution according to the factor applied. A heat treatment always induces an increase for the 6 flavonoids studied and always higher than those corresponding to a light exposure. Therefore, the degradation pathways are different according to the factor applied thus leading to different degradation products from the same native flavonoid [6,7]. Degradation products of rutin, eriodictyol, and quercetin after a thermal treatment and an exposure to light have been found different. For example, polymers of rutin are obtained by the effect of the temperature whereas a cleavage of the molecule occurs after an exposure to light for several days. Moreover, heat treatment of eriodictyol, improved the cellular antioxidant activity, certainly due to the formation of three new products P1 (3-(3, 4-dihydroxyphenyl)-3-hydroxypropanoic acid, P2 (3-(3, 4-dihydroxyphenyl) propanal) and P3 (unidentified product) [41,42].

Comparison of the ΔTEAC results obtained by a heat treatment revealed that the ΔTEAC evolution of eriodictyol, luteolin 7-*O*-glucoside, and rutin have the same profile. ΔTEAC increases until 60 min for eriodictyol and luteolin 7-*O*-glucoside and 45 min for rutin, then decreases. One assumption to account for this phenomenon is that degradation products obtained are themselves degraded in second degradation products. We can notice that these 3 molecules are the flavonoids having the highest number of hydroxyl groups.

Mesquitol have also a high number of phenol groups but it possesses a hydroxyl group in position 3 which gives to the molecule a high reactivity. Indeed, mesquitol has been shown to be very sensitive to heat and to light. Thus, its high reactivity will be interesting to exploit because high antioxidant activities were obtained after degradation. For example, an increase of ΔTEAC reach 200 mM after 120 min of a heat treatment at 130 °C and 40 mM with 20 h of light exposure. Among the other flavonoids, naringin is not very sensitive to heat, only 20% is degraded at a temperature of 130 °C

during two hours. Despite this, its ΔTEAC is equal to 10%. Stronger heat processing must be applied to naringin model solutions to check antioxidant potential of the degradation products of naringin.

Conclusions on the evolution of ΔTEAC of the six flavonoids after a light exposure are difficult to draw. Indeed, except for rutin and mesquitol, ΔTEAC are low and standard deviations on measurements high. Only an exposure to light of eriodictyol and luteolin 7-*O*-glucoside solutions led to prooxidant degradation products.

## 5. Conclusions

Studying model solutions of flavonoids is essential in order to avoid interactions with other molecules. Thus, the simple effect of factors such as temperature and light can be observed. The application of processes on food matrices containing flavonoids is complex because, as it has been shown, they induce changes in antioxidant activities. These evolutions are different depending on the flavonoid and the process applied. A degradation of flavonoids by exposure to light in the food processes does not seem interesting as the gain in antioxidant activity is not high enough. However, heat processing of the flavonoids could be an interesting phenomenon as it induces an increase of the in vitro and in vivo antioxidant activities without causing cytotoxicity of the degradation products obtained.

**Author Contributions:** Conceptualization, I.I., L.C. and M.G.; supervision, I.I., L.C. and M.G.; writing—original draft, I.I. and L.C.; writing—review & editing, I.I. All authors have read and agreed to the published version of the manuscript.

**Funding:** This research received no external funding

**Conflicts of Interest:** The authors declare no conflict of interest

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
