# Peer review of "Effect of Heat Treatment and Light Exposure on the Antioxidant Activity of Flavonoids"

_processes, doi:10.3390/pr8091078_

Round 1

Reviewer 1 Report

The manuscript presented by I. Ioannou et el. describes the influence of heat and light on the flavonoid activity.The work is well conducted and the results are in a way expected.

The authors should explain why they choose 130 oC as limit of heating temperature, maybe it is sufficient just to measure the antioxidant activity up to 100 oC.

What represent the a,b superscript in line 100?

In the main text a new parameter to follow the evolution of the antioxidant activity is proposed. I believe that is a good idea, and an interesting argumentation is provided.

Conclusion remarks should be gathered into a specific chapter at the end of the manuscript.

In my opinion the manuscript can be accepted for publication after these minor corrections.

Author Response

Reviewer 1

Comments and Suggestions for Authors

The manuscript presented by I. Ioannou et el. describes the influence of heat and light on the flavonoid activity.The work is well conducted and the results are in a way expected.

  1. The authors should explain why they choose 130°C as limit of heating temperature, maybe it is sufficient just to measure the antioxidant activity up to 100°C.

130°C was chosen as the upper limit temperature because we considered that most of the temperatures applied during food processes rarely exceed 130°C. In addition, we found that the repeatability of the degradation kinetics greatly decreases beyond 150°C. A sentence was added l.86-88: “Studying temperatures above 130 °C did not seem relevant as they were not very representative of the temperatures used in food processes”

  1. What represent the a,b superscript in line 100?

The superscripts make it possible to differentiate two publications from the same author written in the same year. However, as it was suggested by the second reviewer, the materials and methods have been expanded and the superscripts have been removed.

In the main text a new parameter to follow the evolution of the antioxidant activity is proposed. I believe that is a good idea, and an interesting argumentation is provided.

  1. Conclusion remarks should be gathered into a specific chapter at the end of the manuscript.

As it was suggested by the reviewer, we added a section conclusion as follows: “ΔTEAC calculated in this paper corresponds to the TEAC induced by the degradation products or by the new synergies of the molecules in the solution. For a given flavonoid, ΔTEAC has a different evolution according to the factor applied due to the different degradation pathways. Structural elements such as glycosylation or enone structure does not seem to have an impact on the evolution of the ΔTEAC of the solution. A degradation of flavonoids by exposure to light in the formulation processes does not seem interesting as the gain in antioxidant activity is not high enough. However, heat processing of the flavonoids could be an interesting phenomenon as it induces an increase of the in vitro and in vivo antioxidant activities without causing cytotoxicity of the degradation products obtained.”

In my opinion the manuscript can be accepted for publication after these minor corrections.

Reviewer 2 Report

Dear Authors, please see comments and suggestions in attached file. 

Author Response

Reviewer 2

Manuscript entitled „Effect of a heat treatment and a light exposure on the antioxidant activity of flavonoids” (Processes-865893) presents interesting research topic, however manuscript needs improvement, which should be addressed before considering for publication in Processes journal. It should be pointed out that Materials and methods section is greatly incomplete; missing part dedicated statistical/data analysis (this information was partially inserted below tables or figures), please correct accordingly. Also, I strongly recommend giving the text to a native speaker, extensive editing of English language and style required. All manuscript should be corrected accordingly.

As I see, presented study is another work of Authors connected with the topic of flavonoids. Authors previously focused also on effect of light, oxygen and heat treatment on the evaluation of antioxidant activity, so it proves the Authors' experience in this research field.

  1. Please also explain why Authors decided to study once again just these six-model flavonoid solutions? Also, please carefully explain the novelty of this research in view of other papers published by Authors (e.g. Effect of heat processing on thermal stability and antioxidant activity of six flavonoids published in Journal of food processing and preservation, but also other papers cited). Please demonstrate the differences, because I need to admit that I see lots of similarities.

We decided to study the same 6 flavonoids because we had useful information on these molecules which allowed us to save time in our experiments such as the method of solubility and the solubility limit of each flavonoid, the dilution ranges for the use of the ABTS method. Moreover, the diversity of structure between these 6 flavonoids is of great interest for establishing links between the evolution of the anti-oxidant activity and the presence of a particular structure (catechol or enone structure, glycosylation,…).

Previously, we published a paper on the effect of temperature on the flavonoid degradation with a complete kinetic study (constant rate according to the temperature, activation energy according to the flavonoid structure and the identification of degradation products). We have also published the same kind of study on the effect of light and oxygen. However, by crossing the results of these 2 publications, we realized that important conclusions appeared, this is why we decided to write this new publication. Thus the objective of this publication is to focus on the evolution of the antioxidant activity during the application of change of temperature or light exposure. We have shown that the degradation pathways depend on the factor applied and that certain molecules degrade positively, ie with a significant increase in their antioxidant activity. These conclusions could not be made by the two other publications because the light and temperature factors were studied separately. In addition, the part on in vivo biological activities makes it possible to show that the degradation of the flavonoid solution does not cause cytotoxicity and does indeed bring a positive effect on living cells. One important conclusion to remember from this article is that heat treatment during food formulation processes is not always harmful for antioxidants.

I have following comments which should be addressed before considering this manuscript for publication.

  1. Title seems to be ok, but it suggests some correction as follows: Effect of heat treatment and light exposure on the antioxidant activity (or may be better potential) of flavonoids.

As it was suggested by the reviewer, the title was corrected as follows: “Effect of heat treatment and light exposure on the antioxidant activity of flavonoids”

  1. Abstract

Line 13: What Authors understand as “formulation methods”? It is not clear. Please correct here and through the text.

The term “formulation methods” is not correct, the good term is “formulation processes”. In the field of food engineering it includes all processes to prepare finished products from raw materials (mixing , heating,…).

Line 14: Delete “In this article, we study”.

Line 16: In the end of sentence please add “was studied”.

Line 18-19: Please change this sentence “The results obtained…”

The abstract was modified according to the remarks of the reviewer as follows:

“The application of formulation processes can lead to a modification of both the structure and the activities of flavonoids. In this article, the effect of heat treatment and exposure to light on the antioxidant activity of 6 model flavonoid solutions (rutin, naringin, eriodictyol, mesquitol, luteolin and luteolin 7-O glucoside) was studied. The evolution of the antioxidant activity measured after a heat treatment of 130 °C for 2 hours and an exposure to visible light for 2 weeks is measured by the ABTS method. Temperature and light lead to different degradation pathways. Heat treatment tends to lead to an increase in antioxidant activity. In vivo measurements have shown the non-toxicity of these heat-treated solutions and the increase in biological activities for naringin, erodictyol and luteolin 7-O glucoside.”

  1. Keywords should be redrafted, delete “model solution of”, “formulation processes”, change to “light exposure”, and maybe add some 1-2 more (up to 10 as in guide for Authors). The mean role of keywords is expanding range of published paper, so it is recommended that keywords should be a little different than word used in title of manuscript.

As it was suggested by the reviewer, the keywords were modified as follows:” TEAC, flavonoids, heat process, light exposure, degradation products, bioactivities”.

  1. Introduction

Line 29: Can be supplemented with examples of health beneficial effects.

Examples of health benefits have been cited. Cardiovascular or neurodegenerative diseases are the most recognized health effects.

Line 32-33: Rewrite this sentence “To have strong…”

As it was suggested by the reviewer, the sentence was replaced by “The activity of antioxidants is linked to their capacity to scavenge radical species. The configuration and the number of phenoxy groups are an important parameter to improve antioxidant activity.”

Line 41-42: Correct sentence. Authors may indicate the e.g. products.

As it was suggested by the reviewer, examples of products were added: “However, the consumption of flavonoid-rich products is mainly done through the consumption of processed food products that are issued from the transformation of raw materials by formulation processes and have been mixed with other ingredients such as tomato sauce or vegetables soup”.

Line 44, 47, and others: Environmental conditions? Better write simply temperature, light, pH…

As it was suggested by the reviewer, we deleted environmental conditions:” The study of the evolution of the flavonoid antioxidant activity during a change in temperature or light conditions seems relevant to determine the impact of the formulation processes on the flavonoid bioactivities.”

Line 49: Please insert proper citations.

Citations have not been included in the introduction to comply with the guidelines for a short introduction. We have added tables 3 and 4 in references which contain all the citations allowing to justify this sentence.

Line 55: Aim of study should be as a separate/new paragraph.

As it was suggested by the reviewer, line 53 is the beginning of a new paragraph.

Line 57: Which one is new to the Authors knowledge? This is not clear.

The both are new. The sentence was changed as follows: “ In a second part, the evolution of the antioxidant activity of flavonoid model solutions (naringin, eriodictyol, rutin, mesquitol, luteolin and luteolin 7-O glucoside) during the application of an isothermal treatment and an exposure to light are presented and compared.”

Line 67: Change concentrations to “content”.

As it was suggested by the reviewer, “concentrations” was be replaced by “contents”

In my opinion English language and style must be improve. Sometimes it is very hard to understand what Authors wanted to explain and present.

An extensive editing of English language was done.

  1. Materials and Methods

Line 84: Put “in vitro” and “in vivo” in italics.

Heat treatment and light exposure methods were done accordingly to some previous study? Please add proper citations.

Lines 97-100: This part is greatly incomplete! Authors mentioned HPLC method, without any explanation, amounts, reagents, chromatographic conditions, equipment. The same is for other mentioned methods like ABTS (should be always ABTS+•) and cellular antioxidant assay. Well-established methods can be briefly described and appropriately cited, please correct accordingly. Also, statistical analysis is missing in this section, please add this subsection and accurately describe.

We agree with the remarks of the reviewer, this section is too short. We extended it to describe accurately all methods used in this paper.

  1. Results, Discussion

Line 102: Authors in aim and here mentioned TEAC measured by the ABTS+• assay “as a new parameter” to understand the evaluation of the antioxidant activity, please explain.

By the ABTS+• assay, we measured TEAC. In section 3.1, we explain why the analysis of the TEAC is not sufficient to analyze the evolution of the antioxidant activity of the treated solutions. It is why, we decide to define a new parameter called ΔTEAC. The definition of ΔTEAC is well explained in the section 3.1 with the example of a rutin solution.

Line 112: Instead of “heat” rewrite as higher temperatures, especially 110 and 130, or similar.

As it was suggested by the reviewer, the sentence was modified as follows: “Figure 2a shows that rutin is sensitive to high temperatures.”

Line 115: Explain what was the differences of this analysis to previous one?

A complete analysis of the kinetic data of the heat effect has already been published with the modeling of the degradation kinetics, the determination of constant rates and of the activation energy for the 6 model flavonoid solutions. In Section 3.1, we take only the essential information to understandably define ΔTEAC.

Lines 123-142: This part should be transferred to materials and methods section.

It seems difficult to present these equations in material and methods without the introductory part and the example given in section 3.1.

Line 165: Information about SD should be given when described statistical/data analysis. Delete from here.

A paragraph statistical analysis was added in section Material and methods to give information on SD.

Line 171: Name those products, indicate which one, give some examples.

A citation was added in which these degradation products have been identified

Line 175: change to “As it is shown…”

The sentence was modified as follows: “As shown in figure 4a, a positive ΔTEAC is observed for the two flavones during a thermal treatment.”

Line 181: Add proper reference add additionally some numerical result.

A sentence (with the TEAC of the two flavonoids and the corresponding reference ) was added l.293 as follows: “Indeed, TEAC of luteolin is equal to 5.56 ± 1.45 whereas that of luteolin 7-O glucoside is equal to 4.21 ± 0.56 [6].”

Lines 255-258: Information regarding statistical analysis should be presented in materials and methods section, not here. Correct.

As it was suggested by the reviewer, all statistical information has been gathered in the material and methods section.

Line 289: Change to: Comparison of the TEACs results obtained by a heat treatment revealed that the.

The correction has been made.

Line 300: Change “few” to other expression.

The sentence becomes as follow:” naringin is not very sensitive to heat”

Remember to explain every abbreviation used in the text, please check carefully all manuscript.

Author Contributions: please use as it was mentioned in guide for Authors, not whole name but as initials e.g. I.I.;

  1. Tables and figures

Please reconsider deleting horizontal lines from all the chart field, it makes them less readable. Also, redraft axis legend like e.g. Heating time (min); Light exposure (h); Degree of flavonoid degradation (%).

Fig. 3b 3d 4b: Please reconsider improvement of quality, Authors may change the range of y axis, to make the graph greater and more clarify.

Fig. 6: better use content than consternation.

Fig. 7: Please standardize the font and presentation of each graph.

Table 3: Change to Food/ Model solution of flavonoids, because Authors mainly mentioned food. Change Operating conditions of the process to “process parameters”, results of AA change to Antioxidant activity. The same for table 4.

Figures and Tables were modified according to the remarks of the reviewer.

  1. Conclusion section (lines 303-313) is not mandatory to this journal, but this presented in present form is poorly written, please correct and add also some numerical results. Correct accordingly.

A conclusion section was added and the sentences were modified as follows: “ΔTEAC calculated in this paper corresponds to the TEAC induced by the degradation products or by the new synergies of the molecules in the solution. For a given flavonoid, ΔTEAC has a different evolution according to the factor applied due to the different degradation pathways. Structural elements such as glycosylation or enone structure does not seem to have an impact on the evolution of the ΔTEAC of the solution. A degradation of flavonoids by exposure to light in the formulation processes does not seem interesting as the gain in antioxidant activity is not high enough. However, heat processing of the flavonoids could be an interesting phenomenon as it induces an increase of the in vitro and in vivo antioxidant activities without causing cytotoxicity of the degradation products obtained.”

  1. Please check carefully all literature items and correct; for more information needed please see the guide for Authors. Like in line 321 between “F. A. and M. I.” should be written together.

We checked that the references were well presented according to the instructions of the journal.

Reviewer 3 Report

An interesting and useful study.

Remarks:

Solutions of different concentrations were studied, depending on the compound. Wouldn’t be better to compare solutions of the same concentration? Hasn’t the concentration affected the sensitivity to temperature and light? For more concentrated colored solutions, light penetration could be limited.

The methods are not described in detail. There are references to previous papers, which, however, do not contain detailed descriptions as well. E. g., Ref. 7 states: “according to the method described by Re et al.[12]with modification for use in microplates” but there is no explanation of the modified assay. The cells applied should be characterized (origin, culture conditions, or isolation). How MDA, SOD and GPx were assayed? In which units SOD activity was expressed (U is not applicable to SOD assays)? How cell proliferation was measured?

The animal experiment should be also described under Materials and Methods.

In which units TEAC and delta TEAC is expressed?

Author Response

Reviewer 3

An interesting and useful study.

Remarks:

  1. Solutions of different concentrations were studied, depending on the compound. Wouldn’t be better to compare solutions of the same concentration? Hasn’t the concentration affected the sensitivity to temperature and light? For more concentrated colored solutions, light penetration could be limited.

We have chosen to work at the limit of solubility of flavonoids in water rather than working at the same concentration. The difference in color of the solutions was not verified by spectrophotometry but the variations seemed very small. In addition, the least sensitive molecules had different concentrations and structural analogies. So we think there is no influence of solution concentration on our results.

  1. The methods are not described in detail. There are references to previous papers, which, however, do not contain detailed descriptions as well. E. g., Ref. 7 states: “according to the method described by Re et al.[12]with modification for use in microplates” but there is no explanation of the modified assay. The cells applied should be characterized (origin, culture conditions, or isolation). How MDA, SOD and GPx were assayed? In which units SOD activity was expressed (U is not applicable to SOD assays)? How cell proliferation was measured? The animal experiment should be also described under Materials and Methods.

We agree with the reviewer, the section material and methods must be extended. We developed each part of this section to answer to the remarks of the reviewer.

  1. In which units TEAC and delta TEAC is expressed?

TEAC and delta TEAC are expressed in mM. This information was added in the section ABTS measurement and on all the graphs.

Round 2

Reviewer 2 Report

In my opinion provided changes and supplementation significantly improved the quality of presented research. Authors revised manuscript accordingly Reviewers comments. However presented manuscript needs some improvements especially in case of editing correction, which should be addressed before considering for publication.

I do not know what has happened during editing/uploading of revised version of manuscript, but especially from line 725 some strange editing errors appeared. It appears that in case of Table 2, 3 and 4 disappeared table form, so results can not be publish with this presentation. Also please double check the editing of all figures, please do not skip any of them. This comments should be corrected accordingly.

Generally, I recommend publishing presented manuscript after these minor corrections. 

Author Response

Maybe there was a problem uploading of the submission platform because in my files, there is no editing problem and the manuscript stops at line 592.
I upload the documents again, if these problems persist, I can send them directly to the editor by email.